# Piezo-Sensitive Fabrics from Carbon Black Containing Conductive Cellulose Fibres for Flexible Pressure Sensors

**DOI:** 10.3390/ma13225150

**Published:** 2020-11-16

**Authors:** Julia Ullrich, Martin Eisenreich, Yvonne Zimmermann, Dominik Mayer, Nina Koehne, Jacqueline F. Tschannett, Amalid Mahmud-Ali, Thomas Bechtold

**Affiliations:** 1Textilforschungsinstitut Thüringen-Vogtland e.V., Zeulenrodaer Straße 42, D-07973 Greiz, Germany; j.ullrich@titv-greiz.de (J.U.); m.eisenreich@titv-greiz.de (M.E.); y.zimmermann@titv-greiz.de (Y.Z.); 2Kelheim Fibres GmbH, Regensburger Straße 109, D-93309 Kelheim, Germany; Dominik.Mayer@kelheim-fibres.com (D.M.); Nina.Koehne@kelheim-fibres.com (N.K.); 3Research Institute of Textile Chemistry and Textile Physics, Leopold-Franzens-University of Innsbruck, Hoechsterstraße 73, A-6850 Dornbirn, Austria; Jacqueline.Tschannett@uibk.ac.at (J.F.T.); Amalid.Mahmud-Ali@uibk.ac.at (A.M.-A.)

**Keywords:** conductive fibres, cellulose fibres, pressure sensor, smart textiles, viscose fibres, carbon black

## Abstract

The design of flexible sensors which can be incorporated in textile structures is of decisive importance for the future development of wearables. In addition to their technical functionality, the materials chosen to construct the sensor should be nontoxic, affordable, and compatible with future recycling. Conductive fibres were produced by incorporation of carbon black into regenerated cellulose fibres. By incorporation of 23 wt.% and 27 wt.% carbon black, the surface resistance of the fibres reduced from 1.3 × 10^10^ Ω·cm for standard viscose fibres to 2.7 × 10^3^ and 475 Ω·cm, respectively. Fibre tenacity reduced to 30–50% of a standard viscose; however, it was sufficient to allow processing of the material in standard textile operations. A fibre blend of the conductive viscose fibres with polyester fibres was used to produce a needle-punched nonwoven material with piezo-electric properties, which was used as a pressure sensor in the very low pressure range of 400–1000 Pa. The durability of the sensor was demonstrated in repetitive load/relaxation cycles. As a regenerated cellulose fibre, the carbon-black-incorporated cellulose fibre is compatible with standard textile processing operations and, thus, will be of high interest as a functional element in future wearables.

## 1. Introduction

The Scientific and Technology Options Assessment Panel of the European Parliament (STOA) identified wearables as one of the 10 technologies which will change our lives with very promising market prospects for wearables, forecasted to increase to USD 150 billion by 2026 [1]. The introduction of sensors and electronic devices into textile products allows integration of additional functionalities and opens access to the market of intelligent products for new applications [2].

The integration of electrical devices into textiles requires development of flexible conductive structures and the availability of sensor systems to translate external physical stimuli into electrical signals [3,4]. The intended application of a product determines selection of a certain sensor principle; thus, dependent on the technical requirements, a wide number of different sensor concepts have been developed for a given physical parameter [5].

A number of principles have been reported in the literature for measurement of pressure, i.e., force per area, e.g., via measurement of electrical capacity [6], resistivity [7,8], or optical effects [9]. Changes in inductance and the corresponding resonant frequency of circuits were also proposed as a principle to operate a pressure sensor [10].

The combination of conductive films and deformable insulating layers permits construction of capacitors with pressure-sensitive capacity [11,12,13]. Similarly, two conductive lines, woven into a fabric or wrapped in ply structure and separated by an elastic material, have been proposed as thread-like capacitive pressure sensors [14,15,16].

In another approach, the piezo-resistive properties of compressible structures were used to build flexible textile-based pressure sensors [17,18]. Space-resolved pressure-sensitive layers have been assembled via a combination of piezo-resistive nonwoven material and conductive elements with the aim to develop pressure sensors in shoes [19]. Other approaches integrated elastic piezo-sensitive layers in sensor pads [20], shoes [21], seats [22,23], sensor mats [24], and carpets [25,26,27].

In many cases, the sensor element in such devices is built from a compressible conductive material which then changes its electrical resistance upon compression through external forces [28]. The conductive structure is often a web fabric containing conductive fibres [29]. Furthermore, carbon-nanotube-coated three-dimensional (3D) spacer textiles or conductive silicones have been proposed as pressure-sensitive flexible structures [30,31,32].

Higher electrical conductivity is required for the electrical connections between sensor and data processor. Thus, metal-based coatings of threads, braided wires, and wrapped yarns with the use of thin metal films are used [3]. These lines should exhibit low electrical resistance; thus, contribution of the electrical connection to the total resistance of a device remains in the dimension of a few ohms.

Different types of conductive fibre-based structures can be applied to build a pressure-sensitive structure, e.g., use of conductive polymers [33], fibre coatings with carbon-based layers [34], or integration of conductive material into fibres to obtain intrinsically conductive fibres. Moreover, formation of carbon foam through carbonisation of melamine foam has been reported as a route to prepare flexible conductive structures [35].

Conductive cellulose fibres can be obtained using a number of techniques, e.g., electroless deposition of metal layers on the fibre surface or incorporation of a conductive material, e.g., graphite, carbon black (CB), or carbon nanotubes, into the fibre matrix during fibre production [36,37]. In the viscose process, cellulose at first is steeped in concentrated NaOH solution. The formed alkali cellulose is then reacted with carbon disulphide to form the alkali-soluble cellulose xanthogenate. The aqueous alkaline solution of the cellulose xanthogenate is then spun into a coagulation bath containing a mixture of Na_2_SO_4_ and H_2_SO_4_. In the acidic coagulation bath, at first, coagulation of the xanthogenate occurs, then hydrolysis of the xanthogenate into cellulose and carbon disulphide takes place. The chemical inertness of CB makes this material favourable for incorporation during the viscose fibre formation, as the rather harsh chemical conditions applied during the viscose fibre process cause surface corrosion of metal particles such as Ag and Cu.

In addition to chemical inertness during the process of viscose fibre formation, the conductive additives must not disturb the process of fibre spinning through agglomeration and formation of larger particle structures, which then clog the bores of the spinneret [37].

CB is technically used as a pigment for printing and paint formulation. This nontoxic material is available in bulk amounts and at low costs. Thus, research on the incorporation of CB in viscose fibres for production of conductive cellulose is of high interest to elaborate the fundamentals of a scalable and commercially viable technical process. In this study, the formation of conductive viscose fibres through incorporation of CB was studied as a function of added CB. The fibres were characterised by scanning electron microscopy, conductivity measurement, and determination of fibre strength. The conductive viscose fibres were processed into fibre webs to obtain plane piezo-sensitive layers, which were characterised in static and cyclic load experiments for their functionality as pressure sensors.

## 2. Experimental

### 2.1. Preparation and Characterisation of Conductive Viscose Fibres

As a first step, an aqueous dispersion of 20 wt.% carbon black (CB, low structure, regular colour furnace (RCG), average particle size 27 nm; Printex 300, Orion Engineered Carbons, Luxemburg) was prepared with use of an anionic surfactant as dispersant (sodium lignosulphonate). The dispersion was then added to the standard spinning dope (10 wt.% cellulose) to obtain spinning dopes with 3.2, 10, and 30 wt.% CB. Laboratory spinning devices and a pilot-scale spinning unit (both Kelheim Fibres, Kelheim, Germany) were used for viscose fibre production. The viscose dope was filtered and spun to viscose fibres with fineness of 1.7 or 3.3 dtex.

The actual content of CB in the viscose fibres was determined by photometry (double-beam spectrophotometer, Perkin Elmer Lambda 25, Rodgau, Germany).

Laser scanning microscopy of CB-incorporated viscose fibres was undertaken with a laser scanning 3D microscope (VK-X100 series LSM 3D Profile Measurement, KEYENCE, Tokyo, Japan).

Scanning electron microscope photos (SEM) of the fibres were taken with use of a DSM 940A electron microscope (Zeiss, Oberkochen, Germany).

Fibre tenacity and elongation to break were measured using a single-fibre tensile strength tester on the basis of DIN EN ISO 5079 (Fafegraph HR in combination with Vibromat ME, Textechno, Mönchengladbach, Germany). The fineness of the fibres was determined using the vibrational method on the basis of DIN EN ISO 1973 (Vibromat ME, Textechno, Mönchengladbach, Germany).

To characterise fibres in an oriented arrangement, so-called rotor rings were prepared. A mass of 10 g fibres was processed into a parallelised fibre band (Device built by ITV-Denkendorf, Denkendorf, Germany). The surface resistance along the fibre band was measured with a multimeter (Appendix A; Fluke 1587, Glottertal, Germany). To determine the volume resistivity, the fibre rotor ring was packed between two copper plates with 1 cm distance, and the electrical resistance was measured. The fibre conductivity was calculated as the reciprocal value of the measured electrical resistance.

Additionally, the specific surface resistance and the volume resistance of rotor rings were measured using a ring electrode according to DIN EN 1149-1:2006-09. The measurements were undertaken at three different climate conditions (20 °C, 65% relative humidity (RH), 23 °C, 50% RH, and 23 °C, 40% RH). The samples were conditioned in the respective climate for 24 h before measurement. Rotor rings were cut into pieces of 20 mm length, and a mass of 5 g was placed in the ring electrode. A ring electrode with 50.4 mm diameter and a mass of 460 g was used to compress the fibres (Textilelektrode TE 50, H.-P. Fischer Elektronik GmbH&Co, Industrie und Labortechnik KG, Mittenwalde, Germany). The surface and volume resistances were determined with a tera-ohm meter according to DIN EN 1149 (Appendix A; Milli-Tera-Ohmmeter Milli-TO 3, H.-P. Fischer Elektronik GmbH&Co, Industrie und Labortechnik KG).

Fibre samples were processed into yarn using a lab spinning unit (Kelheim fibres, Germany). The determination of the yarn resistance for a length of 10 cm was performed using a textile stripe electrode and a tera-ohm meter (Appendix A; Textilstreifenelektrode TSE 1, H.-P. Fischer Elektronik GmbH&Co, Industrie und Labortechnik KG). Readings were taken after 60 s of equilibration. Results are given as the mean value of five repetitions.

Nonwoven fabrics (mass per area 250 g/m^2^) were prepared through needle punching to obtain a piezo-resistive structure.

### 2.2. Preparation and Characterisation of Piezo-Resistive Nonwovens

A nonwoven material made from 100% viscose fibres (No. 7, 3.3 dtex, 40 mm, 23 wt.% CB) was used to study the piezo-electric behaviour of the material. The plane samples then were placed between two conductive metal foils. The sandwich was placed on a balance, and compression of the web was achieved by stepwise increase of the weight placed on top of the assembly (area 56.8 cm²). A photograph of the set-up is given in Appendix A. The resistance between the top foil and the bottom foil was measured by means of a laboratory multimeter. Three repetitive cycles of a stepwise increase in pressure, followed by stepwise relaxation of pressure, were performed to analyse the recovery after compression. Results are given as the mean value and standard deviation of three independent experiments.

In a next approach, the conductive viscose fibres were blended with synthetic fibres to improve elastic recovery of the nonwoven fabrics. Two different types of nonwoven fabric were prepared with use of a 3.3 dtex viscose fibre (No. 8, 27 wt.% CB content):

Material A contained 50 wt.% CB-incorporated viscose (3.3 dtex, fibre length 40 mm) and 50 wt.% polyester fibre (3.3 dtex, fibre length 60 mm).

Material B contained 65 wt.% CB-incorporated viscose (3.3 dtex, fibre length 40 mm), 30 wt.% polyester fibre (3.3 dtex, fibre length 60 mm), and 5 wt.% polyester bi-component fibre (2.2 dtex, fibre length 51 mm).

For the repetitive load/relaxation cycles, a modified tensile testing unit was used (Zwick Roell Z010). A sandwich structure (100 mm × 100 mm) of two copper plates with a conductive fibre web as the middle structure was mounted in the testing device (Appendix A). A series of 50 load/relaxation cycles was performed, and the change in conductivity as a function of applied pressure was recorded. The cycling was performed within pressure limits of 500 Pa (5 N/100 cm²) and 2300 Pa (230 N/100 cm²).

## 3. Results and Discussion

### 3.1. Fibre Characterisation

The CB was added to the spinning dope in the form of a CB dispersion, which was stabilised by addition of an anionic or nonionic dispersant. The concentration of the conductive material and the state of dispersion of the CB in the fibre determined the final CB content in the fibre and the measured electrical conductivity. The percolation threshold, as well as the fibre conductivity, depended on the amount of conductive additive, the particle size, and distribution in the fibre. Through dispersion with a high shear rate and addition of a dispersing agent, the particle size of the dispersion could be kept below 10 µm, which is a prerequisite to avoid blocking of the spinneret. Immediately before fibre spinning, any larger aggregates of CB were removed by filtration. At the stage of fibre regeneration, weakly bound CB was washed out into the coagulation bath. Thus, the analytically determined amount of CB incorporated in the fibres was lower than the theoretical amount of CB calculated from the addition of CB to the spin dope.

The conductivity of the fibres increased with increasing content of CB; however, as a result of the presence of dispersed CB, a reduction in mechanical properties of the fibres was observed in parallel. The increase in fibre conductivity and the decrease in tenacity and elongation with increasing content of CB are shown in Table 1. A substantial reduction in tenacity was observed at a CB content above 10 wt.%. The threshold for an increase in conductivity was reached at 15 wt.% CB incorporation, which indicates the lower limit for percolation of the CB particles inside the fibre structure (Figure 1).

Fibres with different content in CB and a standard viscose fibre without addition of CB were analysed by laser scanning microscopy. Fibres with 23.1 wt.% CB content were then selected for scanning electron microscopy, to visualise the state of dispersion of the CB in the fibres using a higher resolution. Representative examples are shown in Figure 2 and Figure 3.

When compared to the surface of a standard viscose fibre (Figure 2e) the incorporation of CB led to an increasingly rougher surface (Figure 2a–d). In particular, at high concentration of CB (Figure 2d and Figure 3a), the presence of CB particles could be observed in the photomicrographs. Bigger agglomerates also led to the appearance of bulges at the fibre surface. In viscose fibre spinning, the cellulose fibre was regenerated from a diluted alkaline solution, which contained approximately 10 wt.% cellulose. During fibre regeneration, solid cellulose forms and a substantial shrinkage in cellulose structure occurred. As an estimate, the area of the cross-section was reduced to 10–15% of the initially extruded viscose solution, and the diameter was reduced to one-third. The dimensions of the CB agglomerates in the dope remained constant; thus, visible bulges began to appear at the fibre surface at higher CB content and larger agglomerates appeared at the fibre surface (Figure 2d,e). The diameter of these agglomerates was still substantially smaller than the diameter of the regenerated fibre, as larger agglomerates were already filtered off before the spinning dope passed the spinneret. The presence of a few larger agglomerates of CB can also be observed in the SEM photomicrographs taken with fibres containing 27.2 wt.% CB. Most of the CB, however, was present inside the fibre in highly dispersed form, which was the condition to achieve percolation and electrical conductivity.

The increase in conductivity with CB content and the moisture dependence of the volume resistivity were also studied in measurements of the volume resistance at rotor rings (Figure 4). Volume resistance measured with the rotor rings was used to characterise the conductivity of the fibres as bulk material with fibres packed in low oriented state. Measurements at different relative humidity demonstrated the influence of the ambient conditions on the electrical resistivity. Fibres with low conductivity exhibited a substantial reduction in resistance with increasing relative humidity, while fibres with 23.1 wt.% CB exhibited an increase in resistance at 65% RH.

The electrical resistance of yarn samples was measured to characterise the conductive behaviour of the fibres in a longitudinally oriented arrangement. The lower amount of conductive material in the cross-section of a yarn and the longer distance between the contact points led to high electrical resistance compared to the tests with rotor rings. Again, the resistance of the samples was reduced with increasing relative humidity due to the contribution of absorbed water to the overall conductivity (Figure 5).

All samples exhibited a distinct change in electrical conductivity with a change in relative humidity. The absorption of water into the cellulose structure and on the fibre surface created two effects:Water adsorbed in the cellulose structure and on the fibre surface contributed to the overall conductivity; thus, a reduction in resistivity was observed. The contribution to the conductivity was, however, low; thus, this effect was observed only in fibre assemblies which exhibited a relatively low conductivity. Thus, in the case of fibres assemblies with a volume resistance on the magnitude of gigaohms, the uptake of moisture contributed to the relatively low conductivity.The adsorption of water molecules also led to changes in fibre dimensions and to the formation of molecular layers of water on the fibre surface. These effects could lead to a reduction in conductivity, which was observed only in the case of fibre assemblies with volume conductivity in the dimension of several kiloohms. Here, the uptake of moisture reduced the level of percolation, e.g., through hygral fibre expansion, thus leading to a reduction in conductivity with increasing moisture content (Figure 5).

### 3.2. Needle-Punched Nonwoven Material as a Pressure Sensor

A needle-punched nonwoven material consisting of 100% conductive viscose fibre (sample (7), 3.3 dtex, fibre length 40 mm, 23.1 wt.% CB) was used as a piezo-sensitive layer. A representative example for the compression/relaxation behaviour during three repetitive load/relaxation cycles is given in Figure 6.

The results in Figure 6 demonstrate the pressure sensitivity of the electrical resistance of a needle-punched fibre nonwoven material in the low-pressure region between 200 and 1000 Pa. At a load below 400 Pa, hysteresis between pressure increase and relaxation appeared. During the expansion of the nonwoven material, a higher number of contact points in the nonwoven region and a higher resistance were observed during the phase of relaxation. During the first cycle, fibres in the nonwoven realigned into a more stable structure; thus, the resistance measured during the following load/relaxation cycles stabilised. In the pressure range between 400 and 1000 Pa, a stable relationship and minimal hysteresis between applied pressure and electrical resistance of the nonwoven material were observed.

To improve the load/relaxation behaviour of the nonwoven material, the conductive viscose fibres were blended with more elastic polyester fibres and polyester bicomponent fibres. These fibres contributed to the recovery of the compressed fibre nonwoven material during the relaxation. Two types of samples were studied:Material A (50 wt.% CB-incorporated viscose, 3.3 dtex, fibre length 40 mm and 50 wt.% polyester fibre, 3.3 dtex, fibre length 60 mm)Material B (65 wt.% CB-incorporated viscose, 3.3 dtex, fibre length 40 mm, 30 wt.% polyester fibre, 3.3 dtex, fibre length 60 mm, and 5 wt.% polyester bicomponent fibre, 2.2 dtex, fibre length 51 mm).

The pressure sensitivity of the electrical resistance of the two different nonwoven materials was studied in three repetitive load/relaxation cycles. Results for Material A and Material B are given in Figure 7.

The repeatability and durability of the nonwoven materials under a high number of repetitive load/relaxation cycles were tested using a modified tensile testing unit.

Representative examples for the resistance change during load/relaxation cycles are shown in Figure 8.

Due to the higher content in conductive fibres, a lower resistance was measured with sample B. The presence of crimped bicomponent fibres in sample B also supported the recovery of the compressed structure and, thus, contributed to a higher signal stability in the cyclic tests.

With increasing load, the electrical resistance of samples A and B was reduced to 50 Ω and then increased during the relaxation to the maximum value of 1250 Ω for the Sample B, while, for sample A, higher resistance of 2500–3000 Ω was observed. The improved mechanical stability of sample B was demonstrated with higher reproducibility of the resistance observed at low pressure. Sample A showed a continuous increase in resistance at low pressure during the first 30 cycles. This was an indication of a prolonged phase of fibre reorganisation in the nonwoven structure in a relaxed state, thus leading to a lower number of contact points available for current transport.

The mechanical stability of Material A and Material B led to a rapid recovery of the electrical resistance when the pressure was reduced to the lower limit (Figure 8). The repeatability and durability of the electrical signal over 50 load/relaxation cycles proved the rapid and reproducible recovery of the conductive structure in unloaded stage. Rupture of brittle fibre segments, breakage of conductive fibres, and loss in the number of electrical contacts in the fibre web would lead to a continuous increase in resistance at low pressure. Both nonwoven materials exhibited very stable signals at the upper pressure limit. A more stable electrical resistance at low pressure was measured with Material B.

The sensor pads exhibited a pressure-sensitive resistance on the magnitude of several kiloohms; thus, measurement of the pressure-dependent signal would be possible with the use of standard electronic devices for data processing. The influence of the electrical resistance of the connections on the signal-evaluating device would then be negligible. A higher sensor resistance, e.g., in the dimension of several megaohms would increase the risk of shunt currents in the electrical connections to the sensor, particularly at high humidity or in a wet state. Piezo-resistive structures with very high resistance would, thus, make the experimental extraction of the pressure dependent signal more difficult.

Changes in relative humidity in the ambient atmosphere would be of substantial influence on the electrical resistance; thus, for practical applications, the nonwoven structure should be covered by either coating or wrapping, to avoid any disturbing influence due to climate-dependent moisture sorption or desorption.

## 4. Conclusions

Regenerated cellulose fibres can be modified through incorporation of CB to implement electrical conductivity. Addition of CB into the viscose dope requires formation of stable dispersions and rather high concentrations of CB to achieve percolation. Through the addition of 20 wt.% CB to the viscose dope, a regenerated cellulose fibre with content of 16.4 wt.% CB could be obtained. The conductivity of a standard viscose fibre of 7.7 × 10^−9^ increased to 9.4 × 10^−8^ S/m for the modified fibre. A further increase in CB content to 19.8 wt.% and 23 wt.% increased the conductivity substantially, to 8.8 × 10^−6^ and 0.044 S/m, respectively. The presence of particulate matter in the fibre structure, however, reduced the tenacity of the fibres to 30–50% of the value of a standard fibre.

In experiments to measure conductivity of rotor rings and yarns, a remarkable influence of relative humidity present in ambient air was observed. Sorption of water into fibre assemblies with high electrical resistance, e.g., 10^9^–10^10^ Ω, led to a decrease in resistance. In the case of rotor rings manufactured from fibres with 23.1 wt.% CB, however, the initially low volume resistivity of 10 kΩ increased slightly, most probably due to hygral fibre expansion and adsorption of water onto the fibre surface.

Through a combination of the more rigid CB-containing viscose fibres with elastic polyester fibres, a piezo-sensitive nonwoven fabric was manufactured, which demonstrated pressure sensitivity in the range of very low pressure of 400–1000 Pa. Repetitive load/relaxation cycles demonstrated the repeatability and durability of the sensor mat and the stability of the signal.

The results highlight the potential of CB-incorporated viscose fibres as a cheap functional material for pressure sensor production in smart textile applications. The incorporation of carbon black into the viscose fibres led to a black colour, which limits their application in the visible parts of a garment. However, their use for pressure sensing inside a garment and therapeutic compression textiles, e.g., bandages, in the form of pressure-sensing pads, could be potential applications of the material. The material is of particular interest for sensor design, as the CB-incorporated cellulose fibres are nontoxic, compatible with future recycling, and able to be produced at affordable costs.

The conductive cellulose fibres exhibit high potential for the substitution of nonbiodegradable synthetic material used in other applications and, thus, could become a greener alternative to existing materials used in smart textiles.

## Figures and Tables

**Figure 1 materials-13-05150-f001:**
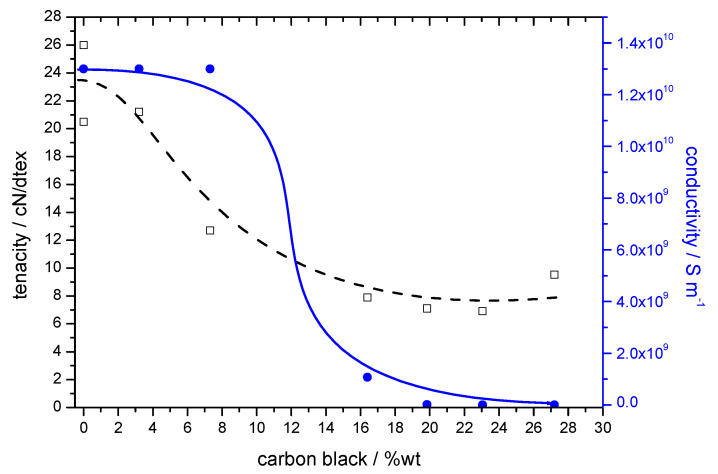
Tenacity and conductivity of viscose fibres as a function of the analytically determined CB content.

**Figure 2 materials-13-05150-f002:**
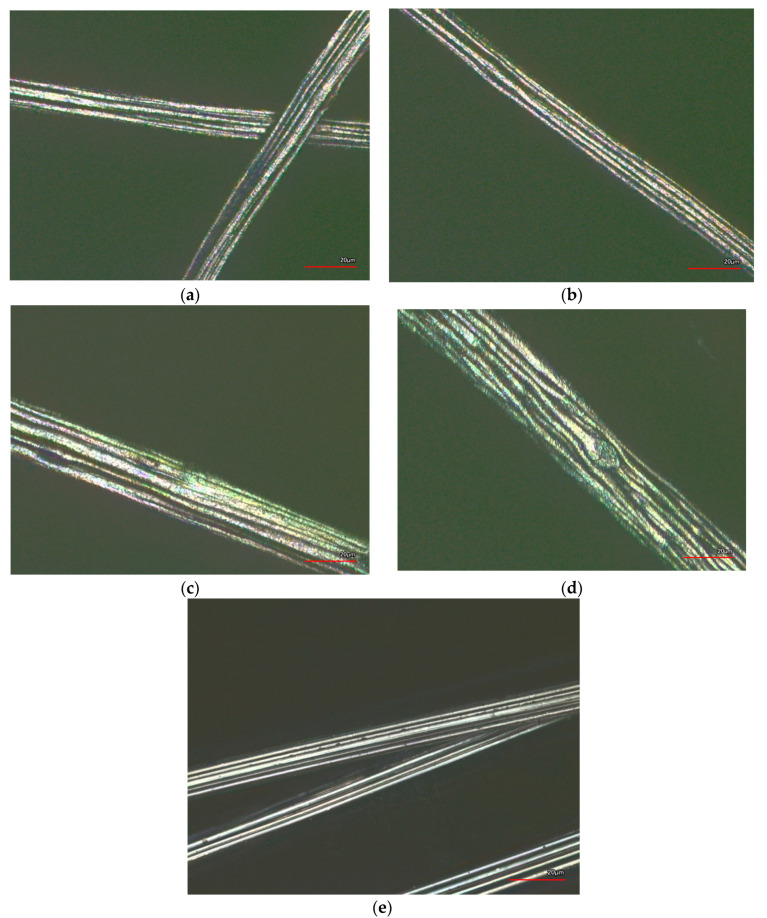
Laser scanning microscopy of CB incorporated viscose fibres: (**a**) sample (4) 1.7 dtex, 7.3 wt.% CB; (**b**) sample (5) 1.7 dtex, 16.4 wt.% CB; (**c**) sample (7) 3.3 dtex, 23.1 wt.% CB; (**d**) sample (8) 3.3 dtex, 27.2 wt.% CB; (**e**) standard viscose fibres 1.3 dtex.

**Figure 3 materials-13-05150-f003:**
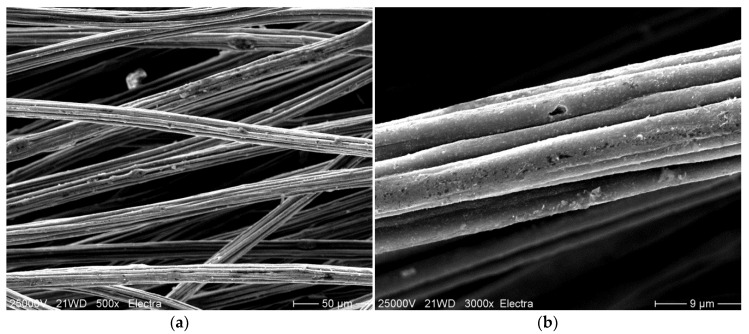
Scanning electron microscopy of CB-incorporated viscose fibres (8) (3.3 dtex, 27.2 wt.% CB); (**a**) magnification 500×, (**b**) magnification 3000×.

**Figure 4 materials-13-05150-f004:**
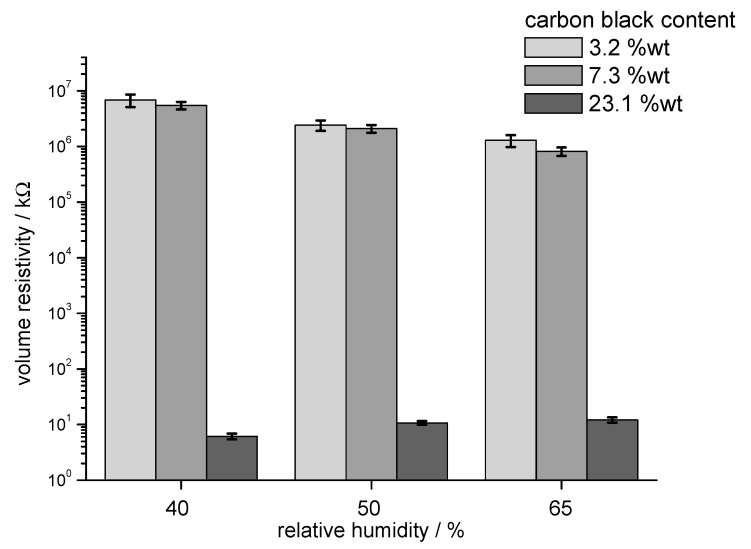
Volume resistivity of rotor rings measured at 40% RH, 50% RH, and 65% RH as a function of incorporated CB: sample (3) 3.2 wt.% CB, sample (4) 7.3 wt.% CB, and sample (7) 23.1 wt.% CB.

**Figure 5 materials-13-05150-f005:**
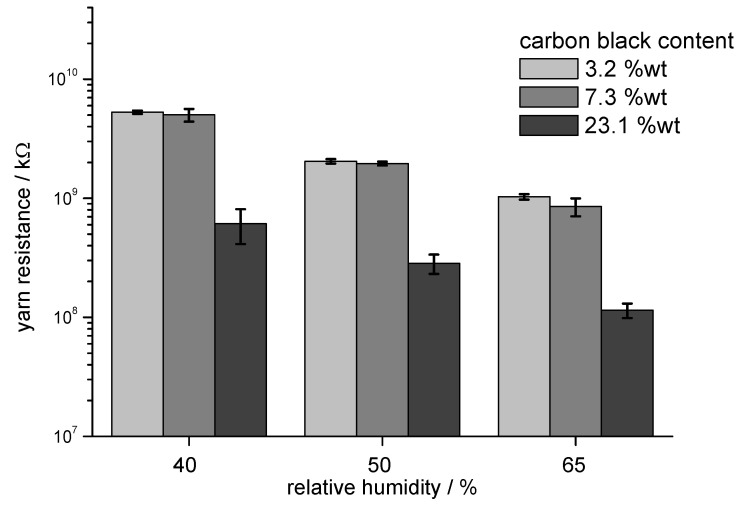
Yarn resistance (measured over a length of 10 cm) measured at 40% RH, 50% RH, and 65% RH as a function of incorporated CB: sample (3) 3.2 wt.% CB, sample (4) 7.3 wt.% CB, and sample (7) 23.1 wt.% CB.

**Figure 6 materials-13-05150-f006:**
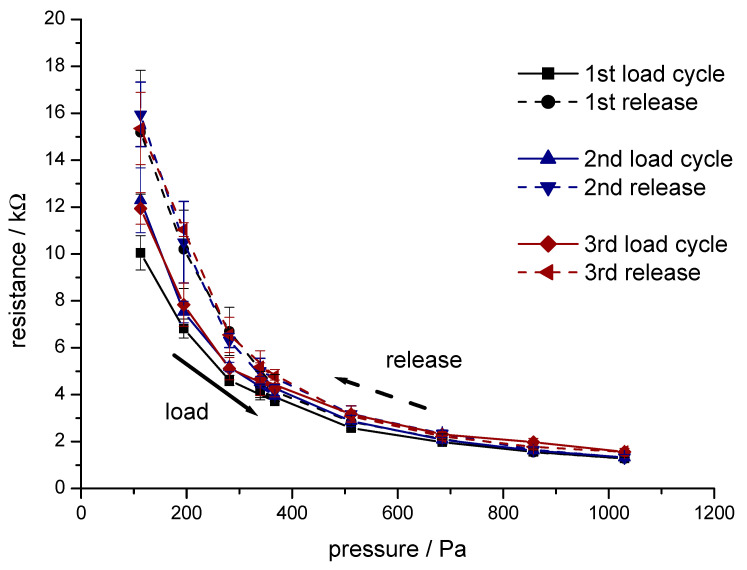
Electrical resistance of a fibre web made of 100% conductive viscose during three repetitive load relaxation cycles (sample (7), 3.3 dtex, fibre length 40 mm, 23.1 wt.% CB, mass per area 250 g/m², test sample area 56.8 cm²).

**Figure 7 materials-13-05150-f007:**
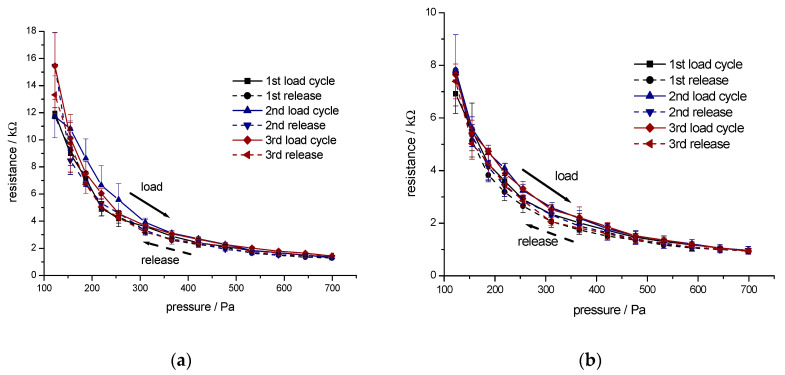
Electrical resistance of a fibre web made of 100% conductive viscose during three repetitive load/relaxation cycles: (**a**) Material A (50 wt.% CB-incorporated viscose, 3.3 dtex, fibre length 40 mm and 50 wt.% polyester fibre, 3.3 dtex, fibre length 60 mm); (**b**) Material B (65 wt.% CB-incorporated viscose, 3.3 dtex, fibre length 40 mm, 30 wt.% polyester fibre, 3.3 dtex, fibre length 60 mm, and 5 wt.% polyester bicomponent fibre, 2.2 dtex, fibre length 51 mm).

**Figure 8 materials-13-05150-f008:**
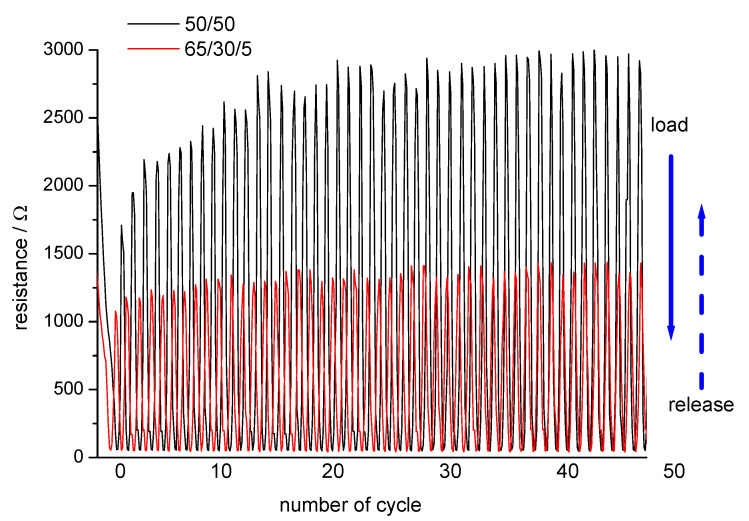
Repeatability and durability of the pressure sensing nonwovens. Resistance change of Material A (50 wt.% CB-incorporated viscose, 3.3 dtex, fibre length 40 mm and 50 wt.% polyester fibre, 3.3 dtex, fibre length 60 mm) shown in black and Material B (65 wt.% CB-incorporated viscose, 3.3 dtex, fibre length 40 mm, 30 wt.% polyester fibre, 3.3 dtex, fibre length 60 mm, and 5 wt.% polyester bicomponent fibre, 2.2 dtex, fibre length 51 mm) shown in red during 50 repetitive load/relaxation cycles.

**Table 1 materials-13-05150-t001:** Mechanical and electrical properties of viscose fibres as a function of carbon black (CB) content added to the spin dope (fibre length 40 mm).

No.	Fibre Fineness	CB Added	CB Incorporated	CB Loss	Tenacity	Elongation	Surface Resistance	Conductivity
	dtex	%	%	%	cN/tex	%	Ω·cm	S/m
1	3.3	0	0	0.00	20.5	24	1.3 × 10^10^	7.7 × 10^−9^
2	1.7	0	0	0.00	26	20	1.3 × 10^10^	7.7 × 10^−9^
3	1.7	3.2	3.2	−0.3	21.2	24.0	1.3 × 10^10^	7.7 × 10^−9^
4	1.7	10	7.3	26.9	12.7	17.4	1.3 × 10^10^	7.7 × 10^−9^
5	3.3	20	16.4	18.0	7.9	15.6	1.1 × 10^9^	9.4 × 10^−8^
6	3.3	25	19.8	20.6	7.1	14.5	11.4 × 10^6^	8.8 × 10^−6^
7	3.3	30	23.1	23.2	6.9	15.2	2.7 × 10^3^	0.044
8	3.3	36.7	27.2	25.9	9.53	30.35	475	0.210

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
