# Peer review of "Piezo-Sensitive Fabrics from Carbon Black Containing Conductive Cellulose Fibres for Flexible Pressure Sensors"

_materials, 2020, doi:10.3390/ma13225150_

Round 1
Reviewer 1 Report
The paper is about use of CNT in the viscose fibers production process. The authors are asked to check the paper with more results and better discussion.
- What is the novelty of this paper? It is obvious that with the increase of CNT to polymers the electrical conductivity would be better and the more CNT, the less resistance would be expected. So, what are authors interested in examining which make this paper different from others and novel. The authors should explain the novelty and goal at the last paragraph of the introduction well.
- What is the viscose dope containing? What device for spinning? The materials and condition should be mentioned.
- The strength is very low with CNT filled viscose fibers. Also, CNT as an nanomaterials in the fibers might be released during the time and also affect the colors. So, what is the application for these textile sensors? The release of the nanomaterials should be checked during time. For how long they keep their performance. Nothing discussed about in the paper. Although researchers cannot examine all the properties in one paper, sth coming to mind at the first look of the paper should be answered for the reader.
- The standards for the all tests should be mentioned in the paper.
- What is the goal of microscopic pictures? The other microscopic pictures from other fibers and also, viscose spun with no CNT would be presented. If it is possible, SEM from cross-section of the fibers to see what is happening with CNT in the fibers.
Author Response
Reviewer 1
The paper is about use of CNT in the viscose fibers production process. The authors are asked to check the paper with more results and better discussion.
- What is the novelty of this paper? It is obvious that with the increase of CNT to polymers the electrical conductivity would be better and the more CNT, the less resistance would be expected. So, what are authors interested in examining which make this paper different from others and novel. The authors should explain the novelty and goal at the last paragraph of the introduction well.
Action: We extended the introduction to explain the relevance and novelty of the process. Lines 86 - 89
- What is the viscose dope containing? What device for spinning? The materials and condition should be mentioned.
Action: We explained the basic steps and chemicals used for the viscose spinning in the introduction. We added details about the spinning units. Lines 75 - 79, lines 198 - 200
- The strength is very low with CNT filled viscose fibers. Also, CNT as an nanomaterials in the fibers might be released during the time and also affect the colors. So, what is the application for these textile sensors? The release of the nanomaterials should be checked during time. For how long they keep their performance. Nothing discussed about in the paper. Although researchers cannot examine all the properties in one paper, sth coming to mind at the first look of the paper should be answered for the reader.
Action: We improved discussion of the newly added Figures 7 and Figure 8 to explain more about durability and repeatability of the signals. We also added some comments in conclusion about application and restrictions due to the black colour of the fibres. Lines 283 - 334, lines 356 - 360
- The standards for the all tests should be mentioned in the paper.
Action: We added the respective standards wherever possible, however some methods were developed specially for low conductive textile materials, line 129
- What is the goal of microscopic pictures? The other microscopic pictures from other fibers and also, viscose spun with no CNT would be presented. If it is possible, SEM from cross-section of the fibers to see what is happening with CNT in the fibers.
Action: We added a LSM picture of a standard viscose fibre without carbon black. Unfortunately we do not have direct access to a SEM. Lines 184, 188 - 198
Reviewer 2 Report
- Please, revise the citation format in the manuscript.
- The authors need to add some information about the fiber spinning process for better understanding for readers.
- Please, describe why both laser scanning microscope and scanning electron microscopy were used for the surface analysis. If it is necessary to use both measurements, the authors need to explain the resulting images in depth.
- The authors need to explain the reason for the comparison of the volume resistance between rotor rings and yarns. There is a lack of discussion in Figure 4 and Figure 5.
- On page 8, if Material B was prepared to improve the load/relaxation behavior of a nonwoven, please, add the result of the same test in Figure 6 with Material B. Typically, Figure 7 shows the repeatability and durability of the sensor, not the load/relaxation behavior.
Author Response
Reviewer 2
- Please, revise the citation format in the manuscript.
Action: We checked and revised the citation format in the manuscript and in the references.
- The authors need to add some information about the fiber spinning process for better understanding for readers.
Action: We added additional information about the spinning process in the introduction section. Lines 74 – 79, lines 198 – 202.
- Please, describe why both laser scanning microscope and scanning electron microscopy were used for the surface analysis. If it is necessary to use both measurements, the authors need to explain the resulting images in depth.
Action: We explained why we used both methods and also added a standard fibre for comparison. We also extended the discussion, lines 188 - 210
- The authors need to explain the reason for the comparison of the volume resistance between rotor rings and yarns. There is a lack of discussion in Figure 4 and Figure 5.
Action: We added a more detailed explanation and discussion of the resistances shown in Figure 4 and 5. Lines 214 – 244.
- On page 8, if Material B was prepared to improve the load/relaxation behavior of a nonwoven, please, add the result of the same test in Figure 6 with Material B. Typically, Figure 7 shows the repeatability and durability of the sensor, not the load/relaxation behavior.
Action: We added the results for the load/relaxation behaviour for Material A and Material B as new Figure 7. Lines 282 - 288
We corrected the explanations for Figure 8 to clarify that the repeatability and durability have been characterised and also revised the conclusions accordingly. Lines 315 – 322.

Reviewer 3 Report
The reviewed paper refers to an interesting and promising issue of the flexible pressure sensor, that may find the utilization not only in wearables but also in other branches of the industry.
In general, the paper is constructed well, with well-analyzed state of the art and well-planned testing programme, except for one issue, which in my opinion is very important.
Authors analyze the influence of humidity showing the changes of volume resistivity with the increment of relative humidity (Fig. 4) and yarn resistance with relative humidity (Fig. 5). Especially the first relation is of great importance for the interpretation of obtained results, and I find it as a very valuable part of the proposed text. Unfortunately, this issue is not developed further; Authors do not refer to that either in point 3.2. Needle punched nonwoven as pressure sensor, nor in the discussion of results and conclusions. They also do not propose any theoretical model for such an interpretation. It is recommended that, in point 3.2. Authors add a tool for interpretation of the results for different humidity levels and summarize it in the discussion of the test results and in the Conclusions. Otherwise, the tests referring to relative humidity find no substantiation in the text.
Additionally, the information in the abstract is unclear – the percolation rate and further – are they obtained from the literature (if so where are the references) or from the research – they find no confirmation in Conclusions – please do comment.
Language and formal issues:
- Use of articles: missing articles, wrong use of articles, unnecessary articles,
- Punctuation errors,
- Mistakes in the use of preposition and determiners,
- Inconsistent spelling – most of the text is written in North American English, but parts of the text in British, e.g. carbonization vs carbonization,
- Inconsistent use of the hyphen, e.g. non-woven vs nonwoven, piezo-resistive vs piezoresistive, please choose one of these options,
- Unclear sense, because of missing words, e.g. Fig. 7. v. 235, 236,
- Lack of spaces between the text and the reference – in the whole text.
The text should be thus carefully checked and corrected.
Author Response
Reviewer 3
Authors analyze the influence of humidity showing the changes of volume resistivity with the increment of relative humidity (Fig. 4) and yarn resistance with relative humidity (Fig. 5). Especially the first relation is of great importance for the interpretation of obtained results, and I find it as a very valuable part of the proposed text. Unfortunately, this issue is not developed further; Authors do not refer to that either in point 3.2. Needle punched nonwoven as pressure sensor, nor in the discussion of results and conclusions. They also do not propose any theoretical model for such an interpretation. It is recommended that, in point 3.2. Authors add a tool for interpretation of the results for different humidity levels and summarize it in the discussion of the test results and in the Conclusions. Otherwise, the tests referring to relative humidity find no substantiation in the text.
Action: We added a detailed discussion and theoretical model for the interpretation of the influence of the humidity on the conductivity in the results and discussion section and in the conclusions. Lines 223 – 244, 331 – 334, 344 - 349
Additionally, the information in the abstract is unclear – the percolation rate and further – are they obtained from the literature (if so where are the references) or from the research – they find no confirmation in Conclusions – please do comment.
Action: We removed the statements about percolation from the abstract.
Language and formal issues:
- Use of articles: missing articles, wrong use of articles, unnecessary articles,
Action: we checked the manuscript and corrected mistakes, lines 30, 40, 166, 168, 186, 305
- Punctuation errors,
Action: we checked the manuscript and corrected mistakes lines 81, 190, 265, 328, 342
- Mistakes in the use of preposition and determiners,
Action: we checked the manuscript and corrected mistakes, lines166, 203, 216, 244
- Inconsistent spelling – most of the text is written in North American English, but parts of the text in British, e.g. carbonization vs carbonization,
Action: We checked and corrected to write the whole text in British English
- Inconsistent use of the hyphen, e.g. non-woven vs nonwoven, piezo-resistive vs piezoresistive, please choose one of these options,
Action: We checked and corrected Lines 54, 150, 351
- Unclear sense, because of missing words, e.g. Fig. 7. v. 235, 236,
Action: We reorganised to bullet points Lines 273 – 277.
- Lack of spaces between the text and the reference – in the whole text.
Action: We corrected throughout the whole text for all references.
The text should be thus carefully checked and corrected.
Action: we checked the manuscript and corrected mistakes Lines 16, 18, 25, 26, 38, 82, 90, Table 1, 182, 203, 289, 306, 338

Round 2
Reviewer 1 Report
no
Reviewer 2 Report
I believe that the previous comments were well addressed in the revised manuscript and it has been improved.
Reviewer 3 Report
Thank you for corrections and explanations.